# Design, Synthesis and Biological Evaluation of Quinazolinamine Derivatives as Breast Cancer Resistance Protein and P-Glycoprotein Inhibitors with Improved Metabolic Stability

**DOI:** 10.3390/biom13020253

**Published:** 2023-01-30

**Authors:** Chao-Yun Cai, Qiu-Xu Teng, Megumi Murakami, Suresh V. Ambudkar, Zhe-Sheng Chen, Vijaya L. Korlipara

**Affiliations:** 1Department of Pharmaceutical Sciences, College of Pharmacy and Health Sciences, St. John’s University, 8000 Utopia Parkway, Queens, New York, NY 11439, USA; 2Laboratory of Cell Biology, Center for Cancer Research, National Cancer Institute, National Institutes of Health, 37 Convent Drive, Bethesda, MD 20892, USA

**Keywords:** quinazolinamine derivatives, multidrug resistance, ABC transporters, BCRP, P-gp

## Abstract

A series of twenty-two quinazolinamine derivatives showing potent inhibitory activities on breast cancer resistance protein (BCRP) and p-glycoprotein (P-gp) were synthesized. A cyclopropyl-containing quinazolinamine **22** was identified as a dual BCRP and P-gp inhibitor, while azide-containing quinazolinamine **33** showed BCRP inhibitory activity. These lead compounds were further investigated in a battery of mechanistic experiments. Compound **22** changed the localization of BCRP and P-gp in cells, thus inhibiting the efflux of anticancer drugs by the two ATP-binding cassette (ABC) transporters. In addition, both **22** and **33** significantly stimulated the ATP hydrolysis of the BCRP transporter, indicating that they can be competitive substrates of the BCRP transporter, and thereby increase the accumulation of mitoxantrone in BCRP-overexpressing H460/MX20 cells. Azide derivative **33**, exhibited a greater inhibitory effect on BCRP after UV activation and can serve as a valuable probe for investigating the interactions of quinazolinamine derivatives with BCRP. Notably, the dual BCRP and P-gp inhibitors **4**–**5**, **22**–**24**, **27**, and BCRP inhibitor **33** showed improved metabolic stability compared to Ko143.

## 1. Introduction

The overexpression of ATP-binding cassette (ABC) transporters is an important cause of multi-drug resistance (MDR) in cancers. ABC transporters with seven subfamilies (ABCA–ABCG) are a superfamily of 48 transporters expressed in humans [1]. Breast cancer resistance protein (BCRP), the second member of the G subfamily of the ABC transporters, also named ABCG2, was identified in an MDR human breast cancer cell line MCF-7/AdrVp in 1998 [2]. Since then, it has been found to be mainly overexpressed in drug-resistant tumors. BCRP is an approximately 75 kDa polytopic plasma membrane protein with one cytoplasmic nucleotide binding domain and one transmembrane domain fused to a single polypeptide chain. BCRP is a half-transporter that dimerizes to become a functional homodimer with a molecular weight of approximately 144 kDa. P-gp, also named ABCB1, is a 141 kDa transporter that contains two transmembrane domains and two cytoplasmic nucleotide binding domains. P-gp is the first human ABC transporter identified through its ability to confer MDR in cancer cells.

The pharmacokinetic behaviors, efficacy, and toxicity of various anticancer drugs are dramatically altered by BCRP and P-gp. Cancers with overexpression of multiple transporters have been reported to be more resistant to chemotherapy than those with a single transporter expression, with worse prognosis. BCRP and P-gp are co-expressed in certain cancers, such as leukemia.

Fumitremorgin C (FTC) (Figure 1) is the first reported BCRP inhibitor [3] with severe neurotoxicity *in vivo*. The FTC analogue Ko143 (Figure 1) was discovered to be a potent BCRP inhibitor with low toxicity [4]. It was reported that Ko143 is not only a BCRP inhibitor but also a weak P-gp inhibitor [5]. Nevertheless, clinical use of Ko143 is hindered due to its rapid metabolism to inactive metabolites in rats [5].

Gefitinib (Figure 2), an epidermal growth factor receptor (EGFR) inhibitor with quinazolinamine moiety, was reported to inhibit BCRP [6] but also modulate the function of P-gp in multidrug-resistant cancer cells [7]. Inspired by previous reports that gefitinib is a BCRP inhibitor [6], Wiese et al. synthesized a series of quinazolinamines (Figure 2) derived from gefitinib as BCRP inhibitors. In addition, the results showed that some quinazolinamines exhibit P-gp inhibitory activities [8,9]. Quinazolinamine derivatives, which share structural similarity with the BCRP and P-gp dual inhibitor gefitinib, have the potential to inhibit both BCRP and P-gp activities. Further investigation of the structure–activity relationship (SAR) of the quinazolinamine derivatives can help discover potent BCRP and P-gp dual inhibitors with enhanced pharmacokinetic properties. The mechanisms of action of quinazolinamine derivatives reversing MDR in cancers were also investigated.

Nitrogen as a hydrogen bond acceptor can play an important role in the interaction of the quinazolinamine derivatives with ABC transporters. Therefore, a nitrogen atom was incorporated, and its position in the aromatic rings varied to investigate the importance of the nitrogen atom and its location. Target compounds with scaffolds A, B, and C (Figure 2) were designed, which contain a nitrogen atom in rings A, B, or C, respectively. The reports showed that quinazolinamine derivatives with the substitutions at *para* or *meta* position are more potent than those with the substitutions at *ortho* position [9]. Both BCRP and P-gp are known to transport hydrophobic compounds [10], suggesting that hydrophobicity plays an important role in the binding affinity of target compounds for these transporters. Thus, methyl, ethyl and n-propyl groups were introduced on rings A or B at *para* position, in scaffolds A (**3**–**5**), B (**12**–**14**), and C (**18**–**20**) to evaluate the role of hydrophobicity on the binding affinity of the target compounds to BCRP and P-gp transporters.

## 2. Materials and Methods

### 2.1. Synthesis of Quinazolinamines

^1^H NMR and ^13^C NMR spectra were acquired with a Bruker 400 Ultrashield^TM^ spectrophotometer (400 MHz). Infrared spectra (IR) were acquired with PerkinElmer Spectrum 100 FT-IR Spectrometers with the exclusive universal ATR accessory. High resolution mass spectra (HRMS) were obtained for all target compounds on a Waters Xevo G2-XS QToF mass spectrometer equipped with H-Class UPLC inlet and a LockSpray electrospray ionization (ESI) source. Reactions were monitored by thin layer chromatography (TLC) and visualized using UV light at 254 nm. TLC was performed using Analtech Uniplate^TM^ Silica Gel GF 250 Micron plates. The purification of the reaction mixtures was conducted using silica gel column chromatography or Reveleris^®^ X2 flash chromatography system by BÜCHI Labortechnik AG. Melting points were determined on a Thomas-Hoover Capillary Melting Point Apparatus. The purity of all target compounds was determined by high-performance liquid chromatography (HPLC), (LC, Agilent 1200 Infinity; column, Agilent HC-C18(2), 170 Å, 4.6 × 250 mm, 5 µm; column temperature, 25 °C; mobile phase, solvent A, methanol, solvent B, water, gradient elution, 30–99% solvent A; flow rate, 1 mL/min; UV signals were recorded at 254 nm). All tested compounds were shown to have >95% purity according to HPLC. Chemicals were purchased from Acros Organics or Alfa Aesar Chemical Company and used without further purification.

2-(Pyridin-4-yl)quinazolin-4(3*H*)-one (**1**). A mixture of anthranilamide (681 mg, 5 mmol), 4-pyridinecarboxaldehyde (536 mg, 5 mmol), iodine (1.40 g, 5.5 mmol), and anhydrous potassium carbonate (690 mg, 5 mmol) in 10 mL of dimethylformamide (DMF) was stirred at reflux for 4 h. Completion of the reaction was monitored by TLC and the mixture was poured into crushed ice to obtain a precipitate. The pH of the mixture was adjusted to 7.0 with concentrated HCl to optimize the precipitation of the desired product. After filtering off the precipitate, it was thoroughly washed with a 20% sodium thiosulfate solution (50 mL) followed by 50 mL of hot distilled water (50 mL). Purification was performed by recrystallization from ethanol to yield 1 as a white solid in 65% yield. ^1^H NMR (400 MHz, DMSO) *δ* 12.83 (s, 1H), 8.80 (dd, *J* = 4.6, 1.5 Hz, 2H), 8.19 (dd, *J* = 7.9, 1.1 Hz, 1H), 8.14–8.09 (m, 2H), 7.93–7.85 (m, 1H), 7.80 (d, *J* = 7.8 Hz, 1H), 7.63–7.55 (m, 1H).

4-Chloro-2-(pyridin-4-yl)quinazolinamine (**2**). Compound **1** (446 mg, 2 mmol) was added to the DMF (5 mL) containing phosphorus oxychloride (0.47 mL, 5 mmol) and stirred for 10 min at room temperature. The mixture was then refluxed for 2 h and the reaction monitored by TLC. After the completion of the reaction, excess phosphorus oxychloride was removed under reduced pressure and the residue was poured into ice water (20 mL). Subsequently, the pH of the mixture was adjusted slowly to 7.0 with 25% NaOH solution and extracted three times with dichloromethane (3 × 20 mL). The organic phase was collected, washed with brine (50 mL) and dried over magnesium sulfate. The solvent was removed under reduced pressure to obtain Compound 2 as a white solid, which was recrystallized from isopropanol in 96% yield. ^1^H NMR (400 MHz, DMSO) *δ* 8.84 (dd, *J* = 4.6, 1.5 Hz, 2H), 8.37 (dd, *J* = 4.5, 1.6 Hz, 3H), 8.28–8.16 (m, 2H), 7.95 (ddd, *J* = 8.2, 6.5, 1.6 Hz, 1H).

#### 2.1.1. General Procedure for the Preparation of the Substituted 4-Anilinoquinazolinamines **3**–**5**

4-Chloroquinazolinamine derivative **2** was added to a solution of a *para* substituted aniline derivative in isopropanol and the mixture was refluxed for a period of 2 h until the completion of the reaction as indicated by TLC. The precipitate that was formed was filtered off, washed with isopropanol (10 mL) and recrystallized from ethanol.

2-(Pyridin-4-yl)-*N*-(*p*-tolyl)quinazolin-4-amine (**3**). Compound **3** was synthesized from Compound **2** (42 mg, 0.17 mmol) reacted with para-methyl aniline (21 mg, 0.20 mmol) as described in the general procedure. It was obtained as a yellow solid (30 mg) in 56% yield, mp. 269–270 °C; IR: 3058, 1562, 1500, 1422, 815, 786 cm^−1^; ^1^H NMR (400 MHz, DMSO) *δ* 10.36 (s, 1H), 9.00 (dd, *J* = 5.3, 1.3 Hz, 2H), 8.72 (d, *J* = 8.3 Hz, 1H), 8.66 (d, *J* = 6.5 Hz, 2H), 8.04–7.95 (m, 2H), 7.82–7.71 (m, 3H), 7.31 (d, *J* = 8.2 Hz, 2H), 2.37 (s, 3H). ^13^C NMR (101 MHz, DMSO) *δ* 158.79, 155.65, 151.17, 149.24, 145.06, 136.26, 134.50, 134.15, 129.52, 128.34, 128.17, 124.34, 123.89, 123.39, 114.84, 21.09. HRMS (ESI) m/z calcd for [C_20_H_16_N_4_ + H]^+^ 313.1453, found 313.1463.

*N*-(4-Ethylphenyl)-2-(pyridin-4-yl)quinazolin-4-amine (**4**). Compound **4** was synthesized from Compound **2** (48 mg, 0.20 mmol) reacted with para-ethyl aniline (26 mg, 0.21 mmol), as described in the general procedure. It was obtained as a yellow solid (50 mg) in 77% yield, **mp**. 228–230 °C; IR: 2969, 1589, 1614, 1366, 798, 776 cm^−1^; ^1^H NMR (400 MHz, DMSO) *δ* 9.96 (s, 1H), 8.76 (d, *J* = 5.8 Hz, 2H), 8.62 (d, *J* = 8.3 Hz, 1H), 8.28 (dd, *J* = 4.5, 1.5 Hz, 2H), 7.92 (d, *J* = 3.8 Hz, 2H), 7.87 (d, *J* = 8.5 Hz, 2H), 7.72–7.65 (m, 1H), 7.33 (d, *J* = 8.5 Hz, 2H), 2.67 (q, *J* = 7.6 Hz, 2H), 1.25 (t, *J* = 7.6 Hz, 3H). ^13^C NMR (101 MHz, DMSO) *δ* 158.63, 157.11, 150.15, 148.96, 148.85, 147.75, 140.12, 136.85, 134.12, 128.59, 128.27, 127.65, 123.59, 123.01, 114.85, 28.15, 16.09. HRMS (ESI) m/z calcd for [C_21_H_18_N_4_ + H]^+^ 327.1610, found 327.1611

*N*-(4-Propylphenyl)-2-(pyridin-4-yl)quinazolin-4-amine (**5**). Compound **5** was synthesized from Compound **2** (241 mg, 1 mmol) reacted with para-propyl aniline (135 mg, 1 mmol), as described in the general procedure as a yellow solid (241 mg) in 71% yield, **mp**. 212–214 °C; IR: 2931, 1513, 1366, 797, 766 cm^−1^; ^1^H NMR (400 MHz, DMSO) *δ* 9.96 (s, 1H), 8.75 (d, *J* = 5.9 Hz, 2H), 8.62 (d, *J* = 8.4 Hz, 1H), 8.28 (d, *J* = 5.9 Hz, 2H), 7.92 (d, *J* = 3.9 Hz, 2H), 7.87 (d, *J* = 8.4 Hz, 2H), 7.69 (dt, *J* = 8.4, 4.1 Hz, 1H), 7.31 (d, *J* = 8.4 Hz, 2H), 2.65–2.57 (m, 2H), 1.65 (dd, *J* = 15.0, 7.5 Hz, 2H), 0.95 (t, *J* = 7.3 Hz, 3H). ^13^C NMR (101 MHz, DMSO) *δ* 158.61, 157.77, 150.65, 150.53, 146.10, 138.35, 137.06, 133.97, 128.85, 128.76, 127.33, 123.52, 122.84, 122.28, 114.87, 37.22, 24.58, 14.13. HRMS (ESI) m/z calcd for [C_22_H_20_N_4_ + H]^+^ 341.1766, found 341.1780.

#### 2.1.2. General Procedure for the Preparation of the Quinazolinamine Derivatives **6**–**8**

A mixture of anthranilamide (1.36 g, 10 mmol), the corresponding aldehyde (10 mmol) and iodine (6.3 g, 25 mmol) in ethanol (20 mL) was refluxed for 6 h. During the reaction, air was pushed into the mixture. Completion of the reaction was monitored by TLC and the mixture was poured into 20% sodium thiosulfate (50 mL) solution followed by hot distilled water (50 mL). Purification was performed by recrystallization from ethanol.

2-(*p*-Tolyl)quinazolin-4(3*H*)-one (**6**). Compound **6** [11] was synthesized as described in the general procedure. A mixture of anthranilamide (681 mg, 5 mmol), the para-methyl aldehyde (661 mg, 5.5 mmol), iodine (1.397 g, 5.5 mmol) in ethanol (20 mL) was refluxed for 5 h. Compound **6** was obtained as a white solid (980 mg) in 83% yield. ^1^H NMR (400 MHz, DMSO) *δ* 12.49 (s, 1H), 8.15 (d, *J* = 7.9 Hz, 1H), 8.11 (d, *J* = 8.2 Hz, 2H), 7.83 (d, *J* = 7.0 Hz, 1H), 7.74 (d, *J* = 8.1 Hz, 1H), 7.52 (t, *J* = 7.6 Hz, 1H), 7.37 (d, *J* = 8.4 Hz, 2H), 2.40 (s, 3H).

2-(4-Ethylphenyl)quinazolin-4(3*H*)-one (**7**). Compound **7** [11] was synthesized as described in the general procedure. A mixture of anthranilamide (680 mg, 5 mmol), the para-ethyl aldehyde (670 mg, 5 mmol), and iodine (1.397 g, 5.5 mmol) in ethanol (30 mL) was refluxed for 7 h. Compound **7** was obtained as a white solid (1.288 g) in 84% yield. ^1^H NMR (400 MHz, DMSO) *δ* 12.50 (s, 1H), 8.15 (dd, *J* = 14.0, 4.8 Hz, 3H), 7.87–7.81 (m, 1H), 7.74 (d, *J* = 8.1 Hz, 1H), 7.55–7.48 (m, 1H), 7.40 (d, *J* = 8.2 Hz, 2H), 2.70 (q, *J* = 7.6 Hz, 2H), 1.23 (t, *J* = 7.6 Hz, 3H).

2-(4-Propylphenyl)quinazolin-4(3H)-one (**8**). Compound **8** [12] was synthesized as described in the general procedure. A mixture of anthranilamide (680 mg, 5 mmol), the para-propyl aldehyde (740 mg, 5 mmol), and iodine (1.397 g, 5.5 mmol) in ethanol (30 mL) was refluxed for 7 h. Compound **8** was obtained as a white solid in 98% yield. ^1^H NMR (400 MHz, DMSO) *δ* 12.49 (s, 1H), 8.18–8.09 (m, 3H), 7.84 (ddd, *J* = 8.6, 7.1, 1.6 Hz, 1H), 7.73 (dd, *J* = 8.2, 0.6 Hz, 1H), 7.55–7.49 (m, 1H), 7.38 (d, *J* = 8.4 Hz, 2H), 2.69–2.62 (m, 2H), 1.72–1.57 (m, 2H), 0.92 (dd, *J* = 8.5, 6.2 Hz, 3H).

#### 2.1.3. General Procedure for the Preparation of the 4-Chloro-quinazolinamine Derivatives **9**–**11**

Quinazolinamine derivative **6**–**8** (2 mmol) was added to the DMF (5 mL) containing phosphorus oxychloride (0.47 mL, 5 mmol) and stirred for 10 min at room temperature. The mixture was then refluxed for 2 h and the reaction monitored by TLC. After completion of the reaction, excess phosphorus oxychloride was removed under reduced pressure and the residue was poured into ice water (20 mL). Subsequently, the pH of the mixture was adjusted slowly to 7.0 with 25% NaOH solution and was extracted three times with dichloromethane (3 × 20 mL). With a separatory funnel, the organic phase was collected, washed with 50 mL brine and dried over magnesium sulfate. The solvent was removed under reduced pressure to obtain a white solid, which was recrystallized from isopropanol.

4-Chloro-2-(*p*-tolyl)quinazolinamine (**9**). Compound **9** was synthesized from Compound 6 as described in the general procedure. Compound **6** (590 mg, 2.5 mmol) reacted with phosphorus oxychloride (5 mL) to obtain a white solid (540 mg) in 85% yield. ^1^H NMR (400 MHz, DMO) *δ* 8.17 (dd, *J* = 7.9, 1.3 Hz, 1H), 8.09 (d, *J* = 8.3 Hz, 2H), 7.87 (ddd, *J* = 8.5, 7.1, 1.5 Hz, 1H), 7.79 (d, *J* = 7.7 Hz, 1H), 7.59–7.52 (m, 1H), 7.39 (d, *J* = 8.0 Hz, 2H), 2.41 (s, 3H).

4-Chloro-2-(4-ethylphenyl)quinazolinamine (**10**). Compound **10** was synthesized with Compound **7,** as described in the general procedure. Compound **7** (500 mg, 2.0 mmol) reacted with phosphorus oxychloride (2 mL) to obtain a white solid (520 mg) in 97% yield. ^1^H NMR (400 MHz, DMSO) *δ* 8.18 (dd, *J* = 7.9, 1.4 Hz, 1H), 8.14–8.08 (m, 2H), 7.92–7.83 (m, 2H), 7.58 (ddd, *J* = 8.1, 6.7, 1.6 Hz, 1H), 7.44 (d, *J* = 8.5 Hz, 2H), 2.72 (q, *J* = 7.6 Hz, 2H), 1.24 (t, *J* = 7.6 Hz, 3H).

4-Chloro-2-(4-propylphenyl)quinazolinamine (**11**). Compound **11** was synthesized with Compound 8 as described in the general procedure. Compound **8** (792 mg, 3.0 mmol) reacted with phosphorus oxychloride (6 mL) to obtain a white solid (770 mg) in 91% yield. ^1^H NMR (400 MHz, DMSO) *δ* 8.42 (d, *J* = 8.3 Hz, 2H), 8.31–8.27 (m, 1H), 8.16–8.09 (m, 2H), 7.84 (ddd, *J* = 8.2, 4.8, 3.3 Hz, 1H), 7.42 (d, *J* = 8.4 Hz, 2H), 2.71–2.63 (m, 2H), 1.73–1.60 (m, 2H), 0.94 (t, *J* = 7.3 Hz, 3H).

#### 2.1.4. General Procedure for the Preparation of the Substituted 4-Anilinoquinazolinamines **12**–**14**

4-Chloroquinazolinamine derivative **9**–**11** (1 equivalent), *p*-aminopyridine (94 mg, 1.1 equivalent) and triethylamine (1.1 equivalent) were taken into isopropanol (5 mL). The mixture was refluxed for a period of 3 h until the completion of the reaction as indicated by TLC. The solvent was removed under reduced pressure and the remaining solid was purified using flash column chromatography.

*N*-(Pyridin-4-yl)-2-(*p*-tolyl)quinazolin-4-amine (**12**). Compound **12** was synthesized as described in the general procedure. Compound **9** (25 mg, 0.1 mmol) was added to a solution of *p*-aminopyridine (10 mg, 0.11 mmol) and triethylamine (11 mg, 0.11 mmol) to obtain a white solid (17 mg) in 53% yield, mp. 278–279 °C; IR: 2997, 1642, 1542, 1370, 1321, 1160, 732 cm^−1^; ^1^H NMR (400 MHz, DMSO) *δ* 9.03 (s, 1H), 8.76 (d, *J* = 7.2 Hz, 2H), 8.47 (d, *J* = 8.2 Hz, 2H), 8.27 (d, *J* = 8.0 Hz, 1H), 8.19 (t, *J* = 7.0 Hz, 1H), 8.01 (d, *J* = 8.2 Hz, 1H), 7.84 (t, *J* = 7.1 Hz, 1H), 7.43 (d, *J* = 8.1 Hz, 2H), 7.13 (d, *J* = 7.1 Hz, 2H), 2.43 (s, 3H). ^13^C NMR (101 MHz, DMSO) *δ* 160.97, 159.50, 159.47, 153.81, 142.80, 142.24, 136.53, 133.79, 130.09, 129.71, 129.11, 128.75, 124.59, 116.78, 109.88, 21.56. HRMS (ESI) m/z calcd for [C_20_H_16_N_4_ + H]^+^ 313.1453, found 313.1455.

2-(4-Ethylphenyl)-*N*-(pyridin-4-yl)quinazolin-4-amine (**13**). Compound **13** was synthesized as described in the general procedure. Compound **10** (134 mg, 0.5 mmol) was added to a solution of *p*-aminopyridine (55 mg, 0.55 mmol) and triethyl-amine (11 mg, 0.11 mmol) to obtain a yellow solid (106 mg) in 65% yield, mp. 234–236 °C; IR: 3294, 2967, 1569, 1503, 826, 755 cm^−1^; ^1^H NMR (400 MHz, DMSO) *δ* 10.11 (s, 1H), 8.59 (dd, *J* = 14.2, 7.3 Hz, 3H), 8.41 (d, *J* = 8.3 Hz, 2H), 8.10 (dd, *J* = 4.9, 1.5 Hz, 2H), 7.93 (d, *J* = 3.8 Hz, 2H), 7.67 (dt, *J* = 8.3, 4.1 Hz, 1H), 7.41 (d, *J* = 8.3 Hz, 2H), 2.71 (q, *J* = 7.6 Hz, 2H), 1.25 (t, *J* = 7.6 Hz, 4H). ^13^C NMR (101 MHz, DMSO) *δ* 159.44, 158.27, 151.17, 150.52, 147.03, 146.98, 136.00, 134.19, 128.68, 128.53, 126.75, 123.55, 115.55, 114.53, 28.55, 15.84. HRMS (ESI) m/z calcd for [C_21_H_18_N_4_ + H]^+^ 327.1610, found 327.1619.

2-(4-Propylphenyl)-*N*-(pyridin-4-yl)quinazolin-4-amine (**14**). Compound **14** was synthesized as described in the general procedure. Compound **11** (177 mg, 0.63 mmol) was added to a solution of *p*-aminopyridine (65 mg, 0.66 mmol) and triethylamine (66 mg, 0.66 mmol) to obtain a yellow solid (134 mg) in 63% yield, mp. 218–220 °C; IR: 2962, 1569, 1504, 822, 754 cm^−1^; ^1^H NMR (400 MHz, DMSO) *δ* 10.12 (s, 1H), 8.60 (dd, *J* = 14.2, 7.3 Hz, 3H), 8.41 (d, *J* = 8.1 Hz, 2H), 8.10 (d, *J* = 6.3 Hz, 2H), 7.93 (d, *J* = 4.3 Hz, 2H), 7.68 (dt, *J* = 8.2, 4.2 Hz, 1H), 7.39 (d, *J* = 8.0 Hz, 2H), 2.70–2.61 (m, 2H), 1.72–1.59 (m, 2H), 0.94 (t, *J* = 7.3 Hz, 3H). ^13^C NMR (101 MHz, DMSO) *δ* 159.45, 158.28, 151.18, 150.53, 146.98, 145.42, 136.03, 134.20, 129.12, 128.68, 128.45, 126.75, 123.56, 115.55, 114.53, 37.58, 24.35, 14.11. HRMS (ESI) m/z calcd for [C_22_H_20_N_4_ + H]^+^ 341.1766, found 341.1773.

#### 2.1.5. Preparation of Compounds **15**–**17**

Methyl 3-benzamidopicolinate (**15**) [13]. To a mixture of methyl 3-aminopicolinate (2.09 g, 5 mmol) and triethylamine (0.7 mL, 5 mmol) in chloroform (10 mL), benzoyl chloride (428 mg, 5.5 mmol) was added dropwise at 5 °C. After stirring at room temperature for 2.5 h, the reaction mixture was diluted with chloroform and washed with saturated sodium bicarbonate (30 mL) and brine (30 mL). The organic layer was dried over sodium sulfate and concentrated in vacuo. The resulting solid was recrystallized from ethyl acetate to obtain Compound 15 as a white solid in 86% yield. ^1^H NMR (400 MHz, DMSO) *δ* 11.32 (s, 1H), 8.74 (dd, *J* = 8.5, 1.5 Hz, 1H), 8.46 (dd, *J* = 4.5, 1.5 Hz, 1H), 8.01–7.95 (m, 2H), 7.75–7.57 (m, 4H), 3.89 (s, 3H).

2-Phenylpyrido[3,2-d]pyrimidin-4(3*H*)-one (**16**) [14]. To a solution of 15 (256 mg, 1 mmol) in methanol (20 mL) was added 28% aqueous ammonia (20 mL). After stirring at room temperature for 2 h, the reaction mixture was filtered to obtain a mixture of uncyclized benzamide. Isopropanol (5 mL) and 2 N sodium hydroxide (2 mL) were added and the crude mixture was heated at reflux for 3 h. The mixture was cooled, neutralized with 2 N HCl, and the solution was evaporated to obtain a precipitate, which was collected to obtain 23 as a white solid in 60% yield. ^1^H NMR (400 MHz, DMSO) *δ* 12.83 (s, 1H), 8.79 (dd, *J* = 4.3, 1.5 Hz, 1H), 8.23–8.13 (m, 3H), 7.84 (dd, *J* = 8.3, 4.3 Hz, 1H), 7.66–7.54 (m, 3H).

4-Chloro-2-phenylpyrido[3,2-d]pyrimidine (**17**). Compound **16** (2 mmol) was added to DMF (5 mL) containing phosphorus oxychloride (0.47 mL, 5 mmol) and stirred for 10 min at room temperature. The mixture was then refluxed for 2 h and the reaction was monitored by TLC. After completion of the reaction, excess phosphorus oxychloride was removed under reduced pressure and the residue was poured into ice water (20 mL). Subsequently, the pH of the mixture was adjusted slowly to 7.0 with 25% NaOH solution and was extracted three times with dichloromethane (3 × 20 mL). The organic phase was collected, washed with brine (50 mL) and dried over magnesium sulfate. The solvent was removed under reduced pressure to obtain a white solid, which was recrystallized from isopropanol in 94% yield. ^1^H NMR (400 MHz, DMSO) *δ* 9.17 (dd, *J* = 4.1, 1.5 Hz, 1H), 8.60–8.47 (m, 3H), 8.13 (dd, *J* = 8.6, 4.1 Hz, 1H), 7.67–7.57 (m, 3H).

#### 2.1.6. General Procedure for the Preparation of the Substituted 4-Anilinoquinazolinamines **18**–**29**

4-Chloroquinazolinamine derivative **2** (1 equivalent) or **17** (1 equivalent) was added to a solution of a para-substituted aniline derivative (1.1 equivalent) and triethylamine (1.1 equivalent) in isopropanol (3 mL) to synthesize **18**–**20**, **22**–**25**, **27**–**29**. In the case of compounds **21** and **26**, 4-dimethylaminopyridine was used in place of triethylamine. The mixture was refluxed for a period of 3 h until completion of the reaction as indicated by TLC. The solvent was removed under reduced pressure and the remaining solid was purified with flash column.

2-Phenyl-*N*-(*p*-tolyl)pyrido[3,2-d]pyrimidin-4-amine (**18**). Compound **18** was synthesized as described in the general procedure. Compound **17** (90 mg, 0.37 mmol) was added to a solution of 4-methylaniline (53 mg, 0.5 mmol) and triethylamine (55.6 mg, 0.55 mmol) to obtain a yellow solid (84 mg) in 73% yield, mp. 161–162 °C; IR: 3333, 1597, 1563, 1409, 802 cm^−1^; ^1^H NMR (400 MHz, DMSO) *δ* 10.11 (s, 1H), 8.59 (dd, *J* = 14.2, 7.3 Hz, 3H), 8.41 (d, *J* = 8.3 Hz, 2H), 8.10 (dd, *J* = 4.9, 1.5 Hz, 2H), 7.93 (d, *J* = 3.8 Hz, 2H), 7.67 (dt, *J* = 8.3, 4.1 Hz, 1H), 7.41 (d, *J* = 8.3 Hz, 2H), 2.71 (q, *J* = 7.6 Hz, 2H), 1.25 (t, *J* = 7.6 Hz, 4H). ^13^C NMR (101 MHz, DMSO) δ 160.29, 157.77, 149.06, 145.60, 138.40, 136.72, 136.41, 133.21, 131.19, 131.12, 129.49, 129.13, 129.00, 128.60, 122.02, 21.03. HRMS (ESI) m/z calcd for [C_20_H_16_N_4_ + H]^+^ 313.1453, found 313.1465.

*N*-(4-Ethylphenyl)-2-phenylpyrido[3,2-d]pyrimidin-4-amine (**19**). Compound **19** was synthesized as described in the general procedure. Compound **17** (90 mg, 0.37 mmol) was added to a solution of 4-ethylaniline (60 mg, 0.5 mmol) and triethylamine (50 mg, 0.5 mmol) to obtain as a yellow solid (121 mg) in 75% yield, mp. 137–138 °C; IR: 3330, 2962, 1595, 1564, 823, 707 cm^−1^; ^1^H NMR (400 MHz, DMSO) *δ* 10.28 (s, 1H), 8.91 (dd, *J* = 4.2, 1.5 Hz, 1H), 8.55–8.45 (m, 2H), 8.28 (dd, *J* = 8.5, 1.5 Hz, 1H), 8.07 (d, *J* = 8.5 Hz, 2H), 7.93 (dd, *J* = 8.5, 4.2 Hz, 1H), 7.61–7.51 (m, 3H), 7.32 (d, *J* = 8.5 Hz, 2H), 2.65 (q, *J* = 7.5 Hz, 2H), 1.23 (t, *J* = 7.6 Hz, 3H). ^13^C NMR (101 MHz, DMSO) *δ* 160.35, 157.72, 149.11, 145.53, 139.69, 138.36, 136.87, 136.36, 131.15, 131.11, 129.14, 129.01, 128.60, 128.33, 121.92, 28.14, 16.16. HRMS (ESI) m/z calcd for [C_21_H_18_N_4_ + H]^+^ 327.1610, found 327.1620.

2-Phenyl-*N*-(4-propylphenyl)pyrido[3,2-d]pyrimidin-4-amine (**20**). Compound **20** was synthesized as described in the general procedure. Compound **17** (77 mg, 0.32 mmol) was added to a solution of 4-propylaniline (47 mg, 0.35 mmol) and triethylamine (39 mg, 0.39 mmol) to obtain a yellow solid (74 mg) in 68% yield, mp. 117–119 °C; IR:3342, 2929, 1594, 1565, 803, 707 cm^−1^; ^1^H NMR (400 MHz, DMSO) *δ* 10.28 (s, 1H), 8.91 (dd, *J* = 4.2, 1.5 Hz, 1H), 8.49 (dd, *J* = 6.6, 3.2 Hz, 2H), 8.29 (dd, *J* = 8.5, 1.5 Hz, 1H), 8.07 (d, *J* = 8.5 Hz, 2H), 7.93 (dd, *J* = 8.5, 4.2 Hz, 1H), 7.56 (dd, *J* = 5.2, 1.8 Hz, 3H), 7.30 (d, *J* = 8.5 Hz, 2H), 2.64–2.56 (m, 2H), 1.70–1.58 (m, 2H), 0.94 (t, *J* = 7.3 Hz, 3H). ^13^C NMR (101 MHz, DMSO) *δ* 160.33, 157.69, 149.06, 145.53, 138.37, 138.00, 136.93, 136.36, 131.12, 129.10, 128.99, 128.89, 128.61, 121.80, 37.22, 24.60, 14.13. HRMS (ESI) m/z calcd for [C_22_H_20_N_4_ + H]^+^ 341.1766, found 341.1775.

*N*-Phenyl-2-(pyridin-4-yl)quinazolin-4-amine (**21**). Compound **21** was synthesized, as described in the general procedure. Compound **2** (24 mg, 0.1 mmol) was added to a solution of phenylamine (10 mg, 0.11 mmol) and 4-dimethylaminopyridine (13 mg, 0.11 mmol) to yield a white solid (25 mg) in 85% yield, mp. 282–284 °C; IR:3261, 3043, 1554, 1523, 1411, 745 cm^−1^; ^1^H NMR (400 MHz, DMSO) *δ* 10.02 (s, 1H), 8.75 (d, *J* = 6.0 Hz, 2H), 8.63 (d, *J* = 8.4 Hz, 1H), 8.28 (d, *J* = 6.0 Hz, 2H), 7.95 (dd, *J* = 9.9, 5.9 Hz, 4H), 7.74–7.66 (m, 1H), 7.50 (t, *J* = 7.9 Hz, 2H), 7.21 (t, *J* = 7.4 Hz, 1H). ^13^C NMR (101 MHz, DMSO) *δ* 158.67, 157.73, 150.72, 150.62, 145.99, 139.44, 134.01, 129.05, 128.82, 127.35, 124.48, 123.62, 122.98, 122.23, 114.91. HRMS (ESI) m/z calcd for [C_19_H_14_N_4_ + H]^+^ 299.1297, found 299.1303.

*N*-(4-Cyclopropylphenyl)-2-(pyridin-4-yl)quinazolin-4-amine (**22**). Compound **22** was synthesized as described in the general procedure. Compound **2** (50 mg, 0.21 mmol) was added to a solution of 4-cyclopropylaniline (42 mg, 0.25 mmol) and triethylamine (25 mg, 0.25 mmol) to obtain a yellow solid (35 mg) in 50% yield, mp. 231–232 °C; IR: 3003, 1515, 1409, 1363, 765 cm^−1^; ^1^H NMR (400 MHz, DMSO) *δ* 9.93 (s, 1H), 8.75 (d, *J* = 5.8 Hz, 2H), 8.60 (d, *J* = 8.4 Hz, 1H), 8.34–8.21 (m, 2H), 7.91 (d, *J* = 3.7 Hz, 2H), 7.83 (d, *J* = 8.5 Hz, 2H), 7.74–7.61 (m, 1H), 7.20 (d, *J* = 8.5 Hz, 2H), 2.03–1.92 (m, 1H), 1.03–0.92 (m, 2H), 0.79–0.67 (m, 2H). ^13^C NMR (101 MHz, DMSO) *δ* 158.56, 157.79, 150.72, 150.58, 146.05, 139.78, 136.80, 133.90, 128.79, 127.24, 125.89, 123.58, 122.89, 122.23, 114.92, 15.26, 9.74. HRMS (ESI) m/z calcd for [C_22_H_18_N_4_ + H]^+^ 339.1610, found 339.1617.

*N*-(4-Isopropylphenyl)-2-(pyridin-4-yl)quinazolin-4-amine (**23**). Compound **23** was synthesized as described in the general procedure. Compound **2** (48 mg, 0.2 mmol) was added to a solution of 4-isopropylaniline (30 mg, 0.22 mmol) and triethylamine (22 mg, 0.22 mmol) to obtain a yellow solid (52 mg) in 76% yield, mp. 225–226 °C; IR: 2958, 1563, 1513, 1415, 764 cm^−1^; ^1^H NMR (400 MHz, DMSO) *δ* 9.95 (s, 1H), 8.84–8.70 (m, 2H), 8.63 (d, *J* = 8.3 Hz, 1H), 8.37–8.20 (m, 2H), 7.91 (t, *J* = 5.7 Hz, 4H), 7.69 (dt, *J* = 8.3, 4.2 Hz, 1H), 7.36 (d, *J* = 8.5 Hz, 2H), 2.95 (dt, *J* = 13.5, 6.7 Hz, 1H), 1.27 (s, 3H), 1.26 (s, 3H). ^13^C NMR (101 MHz, DMSO) *δ* 158.55, 157.81, 150.73, 150.57, 146.05, 144.48, 137.24, 133.91, 128.81, 127.25, 126.77, 123.60, 122.68, 122.24, 114.92, 33.46, 24.49. HRMS (ESI) m/z calcd for [C_22_H_20_N_4_ + H]^+^ 341.1766, found 341.1780.

*N*-(3-Ethylphenyl)-2-(pyridin-4-yl)quinazolin-4-amine (**24**). Compound **24** was synthesized as described in the general procedure. Compound **2** (84 mg, 0.35 mmol) was added to a solution of 3-ethylaniline (48 mg, 0.40 mmol) and triethylamine (40 mg, 0.40 mmol) to obtain a yellow solid (89 mg) in 78% yield, mp. 236–238 °C; IR: 2963, 2440, 1568, 1378, 767 cm^−1^; ^1^H NMR (400 MHz, DMSO) *δ* 10.20 (s, 1H), 8.94 (d, *J* = 4.8 Hz, 2H), 8.68 (d, *J* = 8.1 Hz, 1H), 8.58 (s, 2H), 7.99 (s, 2H), 7.81–7.69 (m, 3H), 7.40 (t, *J* = 7.9 Hz, 1H), 7.09 (d, *J* = 8.0 Hz, 1H), 2.71 (q, *J* = 7.4 Hz, 2H), 1.28 (t, *J* = 7.6 Hz, 3H). ^13^C NMR (101 MHz, DMSO) *δ* 158.95, 155.71, 145.11, 144.67, 138.93, 134.57, 128.99, 128.41, 124.62, 124.36, 124.00, 122.84, 120.86, 114.99, 40.63, 40.42, 40.21, 40.00, 39.80, 39.59, 39.38, 28.77, 16.10. HRMS (ESI) m/z calcd for [C_21_H_18_N_4_ + H]^+^ 327.1610, found 327.1620.

*N*-(3-Propylphenyl)-2-(pyridin-4-yl)quinazolin-4-amine (**25**). Compound **25** was synthesized as described in the general procedure. Compound **2** (31 mg, 0.13 mmol) was added to a solution of 3-propylaniline (21 mg, 0.16 mmol) and triethylamine (39 mg, 0.39 mmol) to obtain a yellow solid (32 mg) in 72% yield, mp. 205–207 °C; IR: 2955, 1524, 1365, 787, 757 cm^−1^; ^1^H NMR (400 MHz, DMSO) *δ* 9.97 (s, 1H), 8.74 (d, *J* = 6.0 Hz, 2H), 8.63 (d, *J* = 8.3 Hz, 1H), 8.29 (d, *J* = 6.0 Hz, 2H), 7.92 (d, *J* = 3.6 Hz, 2H), 7.84 (s, 1H), 7.74 (d, *J* = 8.9 Hz, 1H), 7.72–7.65 (m, 1H), 7.39 (t, *J* = 7.8 Hz, 1H), 7.04 (d, *J* = 7.5 Hz, 1H), 2.68–2.61 (m, 2H), 1.70 (dt, *J* = 14.5, 7.2 Hz, 2H), 0.98 (t, *J* = 7.3 Hz, 3H). ^13^C NMR (101 MHz, DMSO) *δ* 158.65, 157.69, 150.65, 146.05, 142.90, 139.36, 133.97, 128.84, 127.32, 124.65, 123.62, 123.00, 122.23, 120.36, 114.94, 37.94, 24.65, 14.19. HRMS (ESI) m/z calcd for [C_22_H_20_N_4_ + H]^+^ 341.1766, found 341.1779.

2-Nitro-4-((2-(pyridin-4-yl)quinazolin-4-yl)amino)phenol (**26**). Compound **26** was synthesized as described in the general procedure. Compound **2** (24 mg, 0.1 mmol) was added to a solution of 4-amino-2-nitrophenol (17 mg, 0.11 mmol) and 4-dimethylaminopyridine (13 mg, 0.11 mmol) and triethylamine (39 mg, 0.39 mmol) to obtain a yellow solid (30 mg) in 83% yield, mp. 267–268 °C; IR: 3427, 3194, 1527, 1487, 1224, 769 cm^−1^; ^1^H NMR (400 MHz, DMSO) *δ* 8.83–8.69 (m, 2H), 8.43 (d, *J* = 8.3 Hz, 1H), 8.14 (s, 2H), 8.09–7.99 (m, 2H), 7.87 (s, 1H), 7.40 (d, *J* = 5.8 Hz, 2H), 7.06 (d, *J* = 8.9 Hz, 1H), 5.91 (s, 2H). ^13^C NMR (101 MHz, DMSO) *δ* 167.04, 157.17, 152.03, 150.95, 148.30, 144.29, 142.53, 135.93, 134.08, 129.37, 128.50, 126.09, 124.13, 121.88, 120.28, 115.02, 108.96. HRMS (ESI) m/z calcd for [C_19_H_13_N_5_O_3_ + H]^+^ 360.1097, found 360.1103.

*N*-(4-Methoxyphenyl)-2-(pyridin-4-yl)quinazolin-4-amine (**27**). Compound **27** was synthesized as described in the general procedure. Compound **2** (120 mg, 0.5 mmol) was added to a solution of 4-methoxyaniline (68 mg, 0.55 mmol) and triethylamine (55 mg, 0.55 mmol) to obtain a yellow solid (139 mg) in 85% yield. mp. 219–221 °C; IR: 3291, 1510, 1242, 823, 767 cm^−1^; ^1^H NMR (400 MHz, DMSO) *δ* 9.94 (s, 1H), 8.74 (d, *J* = 5.4 Hz, 2H), 8.58 (d, *J* = 8.3 Hz, 1H), 8.26 (d, *J* = 5.4 Hz, 2H), 7.90 (d, *J* = 3.8 Hz, 2H), 7.82 (d, *J* = 8.8 Hz, 2H), 7.67 (dt, *J* = 8.0, 4.1 Hz, 1H), 7.07 (d, *J* = 8.8 Hz, 2H), 3.79 (d, *J* = 22.1 Hz, 3H). ^13^C NMR (101 MHz, DMSO) *δ* 158.64, 157.78, 156.38, 150.64, 150.51, 146.08, 133.75, 132.30, 128.74, 127.10, 124.69, 123.51, 122.21, 114.85, 114.18, 55.72. HRMS (ESI) m/z calcd for [C_20_H_16_N_4_O + H]^+^ 329.1402, found 329.1411.

*N*^1^-(2-(Pyridin-4-yl)quinazolin-4-yl)benzene-1,4-diamine (**28**). Compound **28** was synthesized as described in the general procedure. Compound **2** (58 mg, 0.24 mmol) was added to a solution of 4-aminoaniline (30 mg, 0.28 mmol) and triethylamine (28 mg, 0.28 mmol) to obtain a pale white solid (66 mg) in 88% yield. mp. 301–303 °C; IR: 3445, 3363, 1556, 1512, 1411, 761 cm^−1^; ^1^H NMR (400 MHz, DMSO) *δ* 9.78 (s, 1H), 8.73 (d, *J* = 5.0 Hz, 2H), 8.54 (d, *J* = 8.3 Hz, 1H), 8.24 (d, *J* = 5.6 Hz, 2H), 7.86 (d, *J* = 3.8 Hz, 2H), 7.62 (dt, *J* = 8.1, 4.0 Hz, 1H), 7.49 (d, *J* = 8.6 Hz, 2H), 6.69 (d, *J* = 8.6 Hz, 2H), 5.34 (s, 2H). ^13^C NMR (101 MHz, DMSO) *δ* 158.72, 157.86, 150.57, 150.47, 146.28, 145.79, 133.59, 128.64, 128.13, 126.94, 124.98, 123.45, 122.20, 114.91, 114.34, 40.64, 40.43, 40.22, 40.01, 39.80, 39.59, 39.38. HRMS (ESI) m/z calcd for [C_19_H_15_N_5_ + H]^+^ 314.1406, found 314.1417.

*N*,2-Di(pyridin-4-yl)quinazolin-4-amine (29). Compound **29** was synthesized as described in the general procedure. Compound **2** (20 mg, 0. 08 mmol) was added to a solu-tion of p-aminopyridine (10 mg, 0.09 mmol) and triethylamine (9.1 mg, 0.09 mmol) to obtain a yellow solid (20 mg) in 80% yield. mp. 277–279 °C; IR: 2941, 1645, 1544, 1384, 1171, 776, 745 cm^−1^; ^1^H NMR (400 MHz, DMSO) *δ* 9.07 (s, 1H), 8.87 (d, *J* = 5.9 Hz, 2H), 8.77 (d, *J* = 7.4 Hz, 2H), 8.44 (d, *J* = 6.0 Hz, 2H), 8.36 (d, *J* = 8.4 Hz, 1H), 8.26 (t, *J* = 7.7 Hz, 1H), 8.10 (d, *J* = 8.3 Hz, 1H), 7.95 (t, *J* = 7.6 Hz, 1H), 7.15 (d, *J* = 7.5 Hz, 2H). ^13^C NMR (101 MHz, DMSO) *δ* 161.10, 159.91, 157.58, 153.59, 151.26, 143.73, 142.83, 136.86, 130.84, 129.49, 124.93, 122.36, 117.73, 109.92, 40.63, 40.42, 40.22, 40.01, 39.80, 39.59, 39.38. HRMS (ESI) m/z calcd for [C_18_H_13_N_5_ + H]^+^ 300.1249, found 300.1285.

#### 2.1.7. Preparation of Compounds **30**–**33**

*N*-(4-Isothiocyanatophenyl)-2-(pyridin-4-yl)quinazolin-4-amine (**30**). A solution of amine **28** (31 mg, 0.1 mmol) in dichloromethane (10 mL) was treated dropwise with a solution of di-2-pyridyl thionocarbonate (23 mg, 0.11 mmol) in dicholoromethane (1 mL) over a period of 1 min with vigorous stirring at room temperature. The precipitates were filtered and recrystallized from ethyl acetate to obtain a white solid in 73% yield, mp. 307–309 °C; IR: 3279, 3039, 2053, 1491, 1410, 749 cm^−1^; ^1^H NMR (400 MHz, DMSO) *δ* 10.15 (s, 1H), 8.77 (d, *J* = 5.0 Hz, 2H), 8.61 (d, *J* = 8.4 Hz, 1H), 8.29 (d, *J* = 4.9 Hz, 2H), 8.15–8.02 (m, 2H), 7.95 (d, *J* = 3.4 Hz, 2H), 7.79–7.67 (m, 1H), 7.57 (d, *J* = 8.6 Hz, 2H). ^13^C NMR (101 MHz, DMSO) *δ* 158.39, 157.61, 150.81, 150.71, 145.78, 139.21, 134.24, 133.50, 128.91, 127.56, 126.84, 125.29, 123.65, 123.56, 122.26, 114.94. HRMS (ESI) m/z calcd for [C_20_H_13_N_5_S + H]^+^ 356.0970, found 356.0985.

2-Bromo-*N*-(4-((2-(pyridin-4-yl)quinazolin-4-yl)amino)phenyl)acetamide (**31**). A solution of amine **28** (62 mg, 0.2 mmol) in dichloromethane (2 mL) was treated dropwise with a solution of bromoacetyl bromide (25 mg, 0.2 mmol) in dichloromethane (1 mL) over a period of 1 min with vigorous stirring at 0 °C. The reaction was stirred at room temperature for 24 h. Then, water (10 mL) was added into the mixture, and the precipitates were filtered and recrystallized from ethyl acetate to obtain a yellow solid in 72% yield, mp. 336–338 °C; IR: 3033, 1566, 1510, 763 cm^−1^; ^1^H NMR (400 MHz, DMSO) *δ* 10.49 (s, 1H), 10.15 (s, 1H), 8.87 (d, *J* = 5.2 Hz, 2H), 8.63 (d, *J* = 8.2 Hz, 1H), 8.46 (d, *J* = 4.8 Hz, 2H), 7.95 (d, *J* = 3.8 Hz, 2H), 7.88 (d, *J* = 8.8 Hz, 2H), 7.72 (d, *J* = 8.7 Hz, 3H), 4.08 (s, 2H). ^13^C NMR (101 MHz, TFA) *δ* 169.70, 159.95, 151.90, 147.91, 142.88, 138.31, 138.12, 135.44, 132.91, 131.48, 126.93, 125.42, 123.09, 122.84, 120.14, 112.46, 25.86. HRMS (ESI) m/z calcd for [C_21_H_16_BrN_5_O + H]^+^ 434.0616, found 434.0627.

*N*-(4-((2-(Pyridin-4-yl)quinazolin-4-yl)amino)phenyl)acetamide (32)**.** To a solution of amine **28** (62 mg, 0.2 mmol) and triethylamine (0.28 mL, 2 mmol) in dichloromethane (5 mL), acetic anhydride (0.02 mL, 0.2 mmol) was added. The reaction mixture was stirred at room temperature for 1 h. The precipitates were filtered and recrystallized from ethyl acetate to obtain a white solid in 85% yield, mp. 367–369 °C; IR: 3341, 3038, 1670, 1560, 1510, 760 cm^−1^; ^1^H NMR (400 MHz, DMSO) *δ* 9.98 (d, *J* = 8.7 Hz, 2H), 8.75 (d, *J* = 5.1 Hz, 2H), 8.60 (d, *J* = 8.2 Hz, 1H), 8.27 (d, *J* = 4.9 Hz, 2H), 7.91 (d, *J* = 3.8 Hz, 2H), 7.84 (d, *J* = 8.6 Hz, 2H), 7.69 (d, *J* = 8.8 Hz, 3H), 2.05 (d, *J* = 27.1 Hz, 3H). ^13^C NMR (101 MHz, DMSO) *δ* 168.74, 158.61, 157.78, 150.68, 150.54, 146.06, 136.01, 134.43, 133.95, 128.76, 127.30, 123.56, 123.50, 122.24, 119.60, 114.84, 24.40. HRMS (ESI) m/z calcd for [C_21_H_17_N_5_O + H]^+^ 356.1511, found 356.1521.

*N*-(4-Azidophenyl)-2-(pyridin-4-yl)quinazolin-4-amine (**33**). A solution of amine **28** (62 mg, 0.2 mmol) in water (2 mL) was treated dropwise with conc. hydrochloric acid (0.2 mL) and stirred at 0 °C, followed by dropwise addition of sodium nitrite (14 mg, 0.2 mmol) dissolved in water (1 mL) at 0 °C. A solution of sodium azide (13 mg, 0.2 mmol) in water (1 mL) was then added to the reaction mixture at 0 °C and the reaction mixture was stirred for 2 h at room temperature. After the reaction was stirred at room temperature for 2 h, 5 mL of water was added, and the precipitates were filtered and recrystallized from ethyl acetate to obtain a yellow solid in 59% yield, mp. 304–305 °C; IR: 3324, 3033, 2114, 1505, 1410, 761 cm^−1^; ^1^H NMR (400 MHz, DMSO) *δ* 10.06 (s, 1H), 8.77 (s, 2H), 8.61 (d, *J* = 8.4 Hz, 1H), 8.30 (s, 2H), 8.09–7.97 (m, 2H), 7.94 (d, *J* = 3.5 Hz, 2H), 7.76–7.65 (m, 1H), 7.27 (d, *J* = 8.6 Hz, 2H). ^13^C NMR (101 MHz, DMSO) *δ* 158.55, 157.69, 150.67, 150.60, 145.99, 136.69, 135.02, 134.06, 128.84, 127.41, 124.41, 123.59, 119.76, 114.89. HRMS (ESI) m/z calcd for [C_19_H_13_N_7_ + H]^+^ 340.1311, found 340.1319.

### 2.2. Cell Lines and Cell Culture

The human lung cancer cell line H460, and its MX-selected derivative BCRP-overexpressing cell line, H460/MX20, were used in this study. The KB-C2 cell line was selected by colchicine (Alfa Aesar, Haverhill, MA, USA) (2 μg/mL) with human cervical carcinoma cell line KB-3-1. All the cell lines were maintained in Dulbecco’s Modified Eagle Medium (Corning Inc., New York, NY, USA) containing 10% fetal bovine serum (Gibco Inc., Billings, MT, USA) and 1% penicillin/streptomycin (Gibco Inc.) at 37 °C with 5% CO_2_.

### 2.3. MTT Assay

Cytotoxicity tests and reversal experiments were performed using the 3-(4,5-dimethylthiazol-2-yl)-2,5-diphenyltetrazolium bromide (MTT) colorimetric assay [15]. Cells were seeded evenly into 96-well plates. To determine the cytotoxicity of the quinazolinamine derivatives, each drug at concentrations of 100, 50, 25, 12.5, 6.25, 3.125, 1.56, 0.78 µM were added into the well after 24 h of incubation. To determine the MDR reversal efficacy of the quinazolinamine derivatives, an anticancer drug (mitoxantrone, colchicine, paclitaxel or cisplatin) was added into the designated wells after 2 h pre-incubation with a quinazolinamine derivative or a positive control inhibitor at non-toxic concentrations. After 72 h of drug incubation, the MTT reagent (4 mg/mL) was added into the wells, and then the plates were incubated for an additional 4 h. Subsequently, the supernatant was discarded and 100 µL of dimethyl sulfoxide (DMSO) was added to dissolve the formazan crystals. Cell viability was determined by measuring the absorbance by using an accuSkan™ GO UV/Vis Microplate Spectrophotometer (Fisher Sci., Fair Lawn, NJ, USA) at a wavelength of 570 nm.

### 2.4. Metabolic Stability Study

Human liver microsome (20 mg/mL) 6.25 µL, nicotinamide adenine dinucleotide phosphate (NADPH) (0.75 µmol), MgCl_2_ (0.75 µmol), and the test compound (0.05 µmol) were added into a potassium phosphate buffer (pH 7.4) with 250 µL of final volume [16]. The incubation was carried out aerobically at 37 °C. The mixture was pre-incubated without NADPH for 10 min at 37 °C and NADPH was added to start the reaction. At 1 h after the start of the reaction, an aliquot (50 µL) of the incubation mixture was taken from each incubation and mixed with 150 µL of ice-cold acetonitrile to terminate the reaction. Subsequently, the sample was centrifuged (12,000 rpm) at room temperature. The resulting supernatant was filtered and analyzed using HPLC (LC, Agilent 1200; column, HC-C18(2); column temperature, 25 °C; mobile phase, solvent A, methanol, solvent B, water; gradient elution, 30–99% solvent A; flow rate, 1 mL/min; UV signals were recorded at 254 nm).

### 2.5. Drug Accumulation Assay

To determine the accumulation of drugs on H460, H460/MX20, KB-3-1, and KB-C2 cells, the cells (2.5 × 10^6^ cells/well) were seeded in the 24-well plates and incubated at 37 °C with 5% CO_2_ [17]. After 12 h of incubation, a test compound (5 µM) was added and the plates were incubated at 37 °C for 2 h. Cells were then incubated with 0.01 µM [^3^H]-MX or [^3^H]-paclitaxel-containing medium for an additional 2 h at 37 °C, with or without a test compound. The cells were washed twice with ice-cold phosphate-buffered saline (PBS), trypsinized and lysed at the end of incubation. The radioactivity was measured using the Packard TRICARB1 1900A liquid scintillation analyzer.

### 2.6. Western Blot Assay

Cells in T25 flask were washed with ice-cold PBS. Lysis buffer (100 µL) was added into T25 flask. Using a cell scraper, scrape adherent cells were scraped off the flask and the cell suspension was transferred into a microcentrifuge tube (1.5 mL). Cells were agitated for 20 min at 4 °C and cell lysate was centrifuged mixture at 4 °C for 20 min at 12,000 rpm. The supernatant (lysate) was used for the gel electrophoresis. Equal amounts of total cell lysates (20 µg protein) were resolved by sodium dodecyl sulfate polyacrylamide gel electrophoresis (SDS-PAGE) and electrophoretically transferred onto polyvinylidene fluoride (PVDF) membranes [18]. After incubation in a blocking solution (5% milk) for 2 h at room temperature, the membranes were incubated overnight with primary monoclonal antibodies against GAPDH (GA1R) (Invitrogen, Carlsbad, CA, USA) at 1:1000 dilution of BCRP protein (BXP 21) (Sigma-Aldrich, Inc., St. Louis, MO, USA) (1:1000) or P-gp (F4) (Sigma-Aldrich, Inc., St. Louis, MO, USA) at 4 °C, and were further incubated with horseradish peroxide (HRP)-conjugated secondary antibody (Thermo Fisher Scientific Inc., Waltham, MA, USA) at 1:1000 dilution for 2 h at room temperature. The protein antibody complex was detected using an enhanced chemiluminescence detection system. The grayscale ratio was analyzed by ImageJ and normalized by the grayscale of the ABC protein divided by that of GAPDH.

### 2.7. Immunofluorescence Assay

For immunofluorescence analysis, parental and drug-resistant cells were seeded in 24-well plates at 10,000–20,000 cells/well and incubated for 24 h [19]. The cells were incubated with or without cyclopropyl quinazolinamine **22** or azide quinazolinamine **33** for 24 h, 48 h, and 72 h. Thereafter, cells were washed with PBS and fixed with 4% paraformaldehyde for 10 min at room temperature and then rinsed with PBS twice, followed by permeabilization with 1% Triton X-100 for 10 min at 4 °C. The cells were again washed twice with PBS, and then blocked with 6% BSA for 1 h at 37 °C. Fixed cells were incubated with monoclonal antibody against the BCRP protein (BXP 21) (Sigma-Aldrich, Inc., St. Louis, MO) (1:1000) or P-gp (F4) (Sigma-Aldrich, Inc., St. Louis, MO, USA) overnight at 4 °C, followed by two washes with PBS. The cells were then further incubated with Alexa flour 488 goat anti-mouse IgG (1:1000) (Abcam plc.) for 2 h at 37 °C. After the cells were washed twice with PBS, 4′,6-diamidino-2-phenylindole (DAPI) (2 µg/mL) was used for nuclear counterstaining. The immunofluorescence images were generated using a Nikon TE-2000S fluorescence microscope (Nikon Instruments Inc, Melville, NY, USA).

### 2.8. ATPase Assay

The vanadate (Vi)-sensitive ATPase activity of BCRP and P-gp in the membrane vesicles of High Five insect cells was measured as described previously [20]. The results were presented as vanadate-sensitive ATPase activities by determining the difference in inorganic phosphate liberation measured in the presence and absence of sodium orthovanadate.

### 2.9. Molecular Modeling

Molecular modeling was performed in the Maestro v11.1 (Schrodinger, LLC, New York, NY, USA, 2020) software as described previously [21]. The protein preparation of the wild-type human BCRP (PDB ID: 6FFC) [22] or human P-gp (PDB ID: 7A6E) [23] was performed following the default protocol. The inhibitor was removed; hydrogens were added; the disulfide bonds were created; the waters were removed; restrained minimization was conducted. The grid was generated by selecting residues in the binding pocket of the proteins. The ligands were essentially prepared. The best scoring ligands were obtained through Glide XP docking and used for graphical analysis.

## 3. Results and Discussion

### 3.1. Chemistry

The quinazolinamines **3**–**5** were synthesized via cyclic condensation reaction of amide and aldehyde, chlorination of quinazolinone, and then nucleophilic aromatic substitution (Figure 1A). 2-Substituted quinazolinone **1** was synthesized [8] by a cyclic condensation reaction of anthranilamide with 4-pyridinecarboxaldehyde. Subsequently, the quinazolinone derivative was refluxed with phosphorus oxychloride to obtain 4-chloroquinazolinamine derivative **2**. Nucleophilic aromatic substitution of the 4-chloroquinazolinamine derivative with *para*-substituted anilines was carried out under microwave irradiation to obtain methyl, ethyl, and propyl quinazolinamines **3**–**5** in 56–77% yields, respectively. Compounds **6**–**8** were synthesized via iodine catalysis and air oxidation in 83–98% yields, respectively. Chlorination of quinazolinones **6**–**8** led to compounds **9**–**11**. Nucleophilic aromatic substitution of compounds **9**–**11** led to target quinazoliamine derivatives **12**–**14**.

Compound **16** could not be readily obtained successfully from 3-aminopicolinamide iodine catalysis. Therefore, a two-step synthesis starting with methyl-3-aminopicolinate was used to obtain **16** via Compound **15**, as shown in Figure 1C. Acylation of the amino group of methyl 3-aminopicolinate with benzoyl chloride provided benzamidopicolinate **15**. Treatment of **15** with aqueous ammonium hydroxide followed by cyclization with aqueous sodium hydroxide provided quinazolinone **16** in 60% yield [24]. Chlorination followed by reaction with substituted anilines led to target compounds **18**–**20** in 65–73% yields in 65–73% yields.

Based on the MTT assay results, quinazolinamine derivatives **4** and **5** with scaffold A showed higher potency as BCRP inhibitors than those with scaffolds B or C, decreasing the IC_50_ values of mitoxantrone from 5.66 µM to 0.27 and 0.23 µM, respectively (Table 1). Based on these preliminary results, additional quinazolinamine derivatives with scaffold A (Figure 2) were designed to discover more potent BCRP inhibitors with a potential to also inhibit P-gp. BCRP and P-gp inhibitory activities of all quinazolinamine derivatives were determined to investigate the SAR. Compound **21** with no substitution on ring A was designed for comparison. It was reported that quinazolinamines with substitutions at *meta* and *para* positions were more potent than those with substitutions at the *ortho* position. Two and three carbon substitutions were made at *meta* and *para* positions to obtain target compounds **22**–**25**. The inhibitory activities of these compounds were compared with those of the ethyl- and propyl-substituted compounds **4** and **5**. The 3-NO_2_-4-OH-substituted quinazolinamine has been previously shown to exhibit potent inhibition toward the BCRP transporter [9]. To further investigate the SAR of the quinazolinamine series with scaffold A, the 3-NO_2_-4-OH moiety was introduced in target Compound **26** to probe the effects of the electron withdrawing and donating groups on the reversal activities of quinazolinamines towards BCRP and P-gp transporters. Compounds **27**, **28** and **29** with methoxy, amine and pyridine groups were included to explore the effect of polarity on the inhibitory activities. The quinazolinamine derivatives with electrophilic (**30**–**31**), non-electrophilic control (**32**), or photoaffinity label (**33**) were designed for the investigation of potential covalent bonding between quinazolinamines and ABC transporters.

Target compounds **21**–**29** (Figure 2) were synthesized from Compound **2** in a similar fashion as Compounds **3**–**5**. A nucleophilic aromatic substitution of the 4-chloroquinazolinamine derivatives with various substituted anilines was carried out to obtain the desired quinazolinamines **21**–**29**. Amino derivative **28** was used to synthesize **30**–**33** (Figure 3) by reacting with di(2-pyridyl) thionocarbonate, bromoacetyl bromide, acetic anhydride, sodium nitrite and sodium azide, respectively.

### 3.2. Cytotoxicity and Reversal Study on BCRP

The cytotoxicities of quinazolinamine derivatives were tested to determine the concentration of the quinazolinamines needed for reversal studies. Quinazolinamine derivatives **3**–**5**, **12**–**14**, and **18**–**20** exhibited low to moderate cytotoxicity toward the parental H460 and BCRP-overexperssing H460/MX20 cell lines. The compounds at 5 μM decreased the viability of H460 (Appendix A) and H460/MX20 (Appendix A) cell lines by less than 20%. Next, the reversal assays were conducted on drug resistant H460/MX20 cell line at 5 μM concentration of the quinazolinamine derivatives **3**–**5**, **12**–**14**, and **18**–**20**. Based on the MTT assay (Table 1), ethyl- and propyl-substituted derivatives **4** and **5** exhibited the strongest reversal activities on BCRP-mediated multidrug resistance. Mitoxantrone is a known BCRP substrate. The IC_50_ value of MX on H460 and H460/MX20 are 0.072 μM and 5.66 μM, respectively. When combined with the majority of the quinazolinamine derivatives, the IC_50_ value of MX on H460/MX20 decreased. Ko143, a known BCRP inhibitor, can effectively decrease the IC_50_ value of MX on H460/MX20 to 0.18 μM. The efficacies of Compounds **4** and **5** with Scaffold A are comparable to Ko143, decreasing the IC_50_ values of MX on H460/MX20 to 0.27 μM and 0.23 μM, respectively.

MTT assays on target compounds **21**–**33** were conducted to determine whether these quinazolinamine derivatives at the concentration of 5 μM were non-toxic to H460 (Appendix A) and H460/MX20 cell lines (Appendix A). The results showed that the derivatives at 5 μM did not decrease the viability of H460 and H460/MX20 cells by more than 20%. Thus, the reversal activities of quinazolinamine derivatives were determined at 5 μM. Target compounds **22**–**25** and **27** were found to be most potent with potencies similar to those of **4** or **5** (Table 2). The IC_50_ value of mitoxantrone was found to be 0.42 μM when combined with **33** (5 μM), but upon UV activation of **33** at 254 nm, the IC_50_ value decreased to 0.19 μM. It is hypothesized that the covalent bonding of **33** with BCRP will lead to reduced off-target effects. Gefitinib showed the most potent inhibition on BCRP, decreasing the IC_50_ value of mitoxantrone to 0.13 μM.

### 3.3. Cytotoxicity and Reversal Study on P-gp

Inhibitory activities of the quinazolinamines derivatives **3**–**5**, **12**–**14**, and **18**–**33** on P-gp were studied to investigate the selectivity of these compounds. The quinazolinamine derivatives exhibited low cytotoxicity at the concentration of 5 μM toward the parental KB-3-1 (Appendix A) and P-gp-overexperssing KB-C2 (Appendix A) cell lines. These compounds did not decrease the viability of the KB-3-1 and KB-C2 cells by more than 20%. Hence, the reversal assays were conducted on the drug-resistant KB-C2 cell line with the quinazolinamine derivatives at the concentration of 5 μM. The results indicated that compounds **4**–**5**, **13**, **22**–**25**, and **27** exhibited potent P-gp inhibition (Figure 3). When these quinazolinamines were combined with the P-gp substrate, colchicine (0.5 μM), the survival rate of KB-C2 cells dramatically decreased, compared to the colchicine alone. Therefore, these eight compounds were further investigated, and it was found that compounds **4**–**5**, **22**–**24**, **27** exhibited higher potency than verapamil and gefitinib (Figure 4). The results indicated that in combination with compounds **4**–**5**, **22**–**24**, **27**, the IC_50_ values of colchicine on KB-C2 cells were decreased dramatically from 7.34 μM to lower than 0.30 μM, while verapamil and gefitinib decreased the IC_50_ value of colchicine to 0.43 and 0.55 μM, respectively. In addition, azide derivative **33** did not show potent inhibition on the P-gp with IC_50_ value of 6.05 μM for colchicine after the combination.

### 3.4. Metabolic Stability Study of Quinazolinamine Derivatives

Ko143, as a potent BCRP inhibitor, is widely used as a positive control in the scientific research. However, Ko143 has limitation for clinical use due to the low metabolic stability *in vivo* [25]. Therefore, the design of metabolically more stable reversal agents for BCRP-mediated MDR is highly desired. Human liver microsomes contain a wide variety of membrane-bound drug-metabolizing CYP450 enzymes [26], derived from the human liver, where the drugs are mainly metabolized. Thus, *in vitro* metabolic stability testing with human liver microsomes has been commonly used to predict the *in vivo* hepatic metabolism of the drugs in the human body [27].

The potent BCRP and P-gp dual inhibitors **4**–**5**, **22**–**24**, **27**, and potent BCRP inhibitor **33** were selected for a metabolic stability study. The results showed that 80–92% of the seven quinazolinamine derivatives remained intact after 1 h of incubation with human liver microsomes, indicating higher metabolic stability than Ko143 (Table 3). Thus, these quinazolinamines can serve as valuable pharmacologic tools in the future. Significantly, 92% of cyclopropyl derivative **22** remained after incubation with human liver microsomes compared to 41% of Ko143. It has been reported that the cyclopropyl group can enhance potency, reduce off-target effects, increase metabolic stability, increase brain permeability, decrease plasma clearance, contribute to an entropically more favorable binding to the receptor, restrict the conformation of peptides/peptidomimetics to prevent proteolytic hydrolysis, and alter drug pKa [28]. Since the results showed that **22** is a potent BCRP and P-gp inhibitor with high metabolic stability, it was selected for investigation of its mechanism in reversing the multidrug resistance mediated by BCRP and P-gp. In addition, azide derivative **33,** which was found to be a potent BCRP inhibitor, was selected for further investigation to learn about the binding mode of quinazolinamines to BCRP protein.

### 3.5. The Reversal Study of ***22*** and ***33*** in Combination with Anticancer Drugs

Reversal studies were conducted to further investigate the effects of **22** and **33** in combination with BCRP substrate, mitoxantrone; P-gp substrate, paclitaxel; and non-substrate, cisplatin. The results showed that the reversal effect of **22 (VKCY-1)** on P-gp was not limited to one substrate (Figure 5A). It could sensitize KB-C2 cells to the anticancer drug, paclitaxel, in addition to colchicine. A non-BCRP or P-gp substrate cisplatin, was used as a negative control (Figure 5B,C). The results suggested that the sensitizing effects of Compounds **22 (VKCY-1)** and **33 (VKCY-2)** were limited to BCRP and/or P-gp substrates for ABC-transporter-mediated MDR cancer cell lines.

### 3.6. The Effect of ***22*** and ***33*** on Anticancer Drugs Accumulation

The reversing of multidrug resistance can be achieved through various mechanisms, such as direct binding to P-gp, inhibiting ATPase activity, or altered expression level or localization of ABC transporter proteins. Previous reports indicated that several tyrosine kinase inhibitors (TKIs), such as gefitinib, imatinib nilotinib, and erlotinib, inhibited BCRP-mediated drug resistance. These TKIs themselves are BCRP substrates and may act as competitive BCRP inhibitors to block the efflux of anticancer drugs. In contrast, protein kinase C (PKC) inhibitors can inhibit BCRP or P-gp ATPase activity to suppress drug resistance.

Drug accumulation assays were conducted to determine the effects of cyclopropyl derivative **22** and azide derivative **33** on the accumulation of mitoxantrone in H460 and H460/MX20 cells, or paclitaxel in KB-3-1 and KB-C2 cells. The results of the accumulation assay showed that both cyclopropyl derivative **22** and azide **33** can slightly increase the accumulation of mitoxantrone in parental H460 cells (Figure 6A). Since a low level of BCRP transporter is expressed in H460 cells, **22** and **33** can also inhibit the efflux function of BCRP in H460 cell. In addition, the two compounds increased the accumulation of mitoxantrone in resistant H460/MX20 cells, which overexpressed BCRP transporter. Treatment of **33** followed by UV light activation, showed that the accumulation of mitoxantrone was increased compared to the group without activation. It explained the reason why the IC_50_ value of mitoxantrone for the activated **33** treatment group (Table 2) was lower than that of the inactivated **33** treatment group. In addition, **22** did not increase the accumulation of paclitaxel in the parental cells KB-3-1, while **22** significantly increased the accumulation of paclitaxel in resistant cell line KB-C2 (Figure 6B). When in combination with **22** or **33**, the accumulation of mitoxantrone and paclitaxel in MDR cells were enhanced and the viability of cancer cells decreased. This explained why the IC_50_ values of mitoxantrone and paclitaxel decreased when they were combined with **22** or **33**.

### 3.7. The Effect of ***22*** and ***33*** on BCRP and P-gp Expression Level in Cells

Western blot assays were conducted to study the effects of **22** and **33** on the expression level of BCRP or P-gp proteins. Compound **22** did not change the expression level of BCRP protein on H460/MX20 cells, nor did it change the P-gp protein expression on KB-C2 cells (Figure 7A,B). The BCRP expression level on H460/MX20 cells was not altered by **33** (Figure 7C). Thus, it was concluded that blocking the efflux function of BCRP or P-gp was not due to the downregulation of the expression level.

### 3.8. The Effect of ***22*** and ***33*** on the Localization of BCRP and P-gp

The immunofluorescence assays were conducted to determine the effects of **22** and **33** on the localization of BCRP or P-gp proteins. BCRP protein is expressed on the membranes of drug-resistant cells H460/MX20 and the P-gp protein is expressed on the membranes of KB-C2 cells in the control groups (Figure 8A,B). With the incubation of **22** (5 μM) for 24, 48, and 72 h, the fluorescence on the membranes of H460/MX20 decreased gradually. With the treatment of **22** (5 μM) for 24 and 48 h, the fluorescence on the membranes of KB-C2 cells did not decrease significantly, while the fluorescence decreased slightly at 72 h. Since **22** does not change the expression level of BCRP and P-gp protein based on the Western blot analysis, it was hypothesized that the intracellular localization of the ABC transporters on both H460/MX20 and KB-C2 cells were altered by **22**. The alteration of the ABC transporters’ localization can result in the loss of the efflux function of ABC-transporters in MDR cells. Thus, the anticancer drugs can be accumulated in the cells, and effectively killing them. Interestingly, **33** did not change the intracellular localization of BCRP on H460/MX20 cells (Figure 8C). The reversal effect of **33** on BCRP can be due to other mechanisms instead of the alteration of the BCRP localization.

### 3.9. The Effect of ***22*** on BCRP and P-gp Expression Level at the Plasma Membrane and Intracellular Organelles

To further investigate whether **22** affects the localization of BCRP and P-gp, the proteins were extracted from the membrane and cytoplasm of the cells, respectively. It was found that **22** decreased the expression level of BCRP protein on the membrane of H460/MX20 cells. On the other hand, the BCRP protein expression level in the cytoplasm was increased by target Compound **22** (Figure 9A,B). Compound **22** showed a similar effect on the P-gp protein expression of the KB-C2 cells (Figure 9C,D) that the expression level of the P-gp protein on the membrane slightly decreased and the cytoplasmic P-gp slightly increased through the quantifications with *Image J*. These results revealed that **22** can translocate BCRP or P-gp protein from the membrane to the cytoplasm.

### 3.10. The Effect of ***22*** and ***33*** on BCRP and P-gp ATPase

ATPase assays were conducted to determine whether **22** and **33** affect the BCRP or P-gp ATPase activities. The results of ATPase assay showed that both **22** (Figure 10A) and **33** (Figure 10B) were most likely substrates of the BCRP transporter. Therefore, they could act as potential competitive substrates, thus blocking the efflux of anticancer drugs and increasing the accumulation of anticancer drugs. The results indicated that **22** (Figure 10C) can neither stimulate or inhibit the P-gp ATPase; thus, it is not the substrate or inhibitor of BCRP. From the results of the Western blot and ATPase assay, **22** did not change the P-gp protein expression level and did not affect the P-gp ATPase activity. Therefore, **22** likely reverses the P-gp-mediated MDR by altering the localization of P-gp on MDR cancer cells.

### 3.11. Docking Analysis of ***22*** and ***33*** with BCRP and P-gp

The Glide docking scores of **22**, **33**, and Ko143 binding with BCRP (PDB 6FFC) are −11.123, −10.403 and −12.017 kcal/mol, respectively (Figure 11). The quinazoline ring and the pyridine ring of **22** interacted with the Phe439 of the BCRP model through π-π interactions. The amino group of **22** formed a hydrogen bond with the carbonyl group in the side chain of Asn436 (Figure 11A,D). The quinazoline ring of **33** interacted with the Phe439 of the BCRP model through π-π interactions (Figure 11B,E). The nitrogen in the pyridine ring of **33** formed a hydrogen bond with the amino group in the side chain of Asn436. The amino group of Ko143 interacted with the carbonyl group in the side chain of Asn436 through a hydrogen bond. The carbonyl group of Ko143 formed a hydrogen bond with the hydroxyl group of Thr542. (Figure 11C,F). Compounds **22** and **33** were stabilized into a pocket formed by residues Phe432, Thr435, Thr542, Val546, and Met549 of BCRP. The Glide docking scores of **22** and verapamil binding with P-gp (PDB 7A6E) [23] were −8.288 and −7.774 kcal/mol, respectively (Figure 12). The quinazoline ring of **22** interacted with Phe343 and Trp232 of the P-gp model through π-π interactions. The tertiary amine of verapamil can become a cation in cancers due to the acidic tumor microenvironment [29] and form a salt bridge with the side chain of Glu875 in P-gp.

## 4. Conclusions

In this study, twenty-two quinazolinamine derivatives were synthesized, and their reversal activities for BCRP- and P-gp-mediated MDR were determined. Based on the SAR, quinazolinamines with scaffold A were more potent BCRP inhibitors than those with scaffolds B or C. The results indicated that alkyl substituted quinazolinamine analogues **4**–**5**, **13**, **22**–**25**, and methoxy quinazolinamine **27** are potent BCRP and P-gp dual inhibitors with a potential to reverse MDR by blocking the efflux of anticancer drugs. In addition, selected quinazolinamine derivatives (**4**–**5**, **22**–**24**, **27**, and **33**) that were investigated for metabolic stability exhibited higher metabolic stability than Ko143. Cyclopropyl quinazolinamine **22** potently inhibited both BCRP- and P-gp, significantly increased the accumulation of mitoxantrone in BCRP-overexpressing H460/MX20 cells, or paclitaxel in KB-C2 cells with P-gp overexpression. The results indicated that **22** changed the localization of BCRP in H460/MX20 cells and P-gp in KB-C2 cells but did not alter the expression level of BCRP or P-gp, thus blocking the efflux function of ABC transporters. In addition, the stimulation of ATP hydrolysis of BCRP by **22** can be another mechanism for reversing the BCRP-mediated MDR. Interestingly, compared to **22**, azide quinazolinamine **33** is a BCRP inhibitor with a different mechanism in that it does not alter the localization of BCRP protein. In addition, **33** did not change the BCRP protein expression level. The ATPase results showed that **33** significantly stimulated the ATP hydrolysis of BCRP transporter indicating it could be a competitive substrate of the BCRP transporter to block the efflux of anticancer drugs. Azide quinazolinamine **33** with a photoaffinity label can be activated by UV light and may covalently bind with the BCRP transporter. Thus, it can serve as a probe to investigate the binding site of quinazolinamine derivatives on the BCRP protein. From the molecular-modeling results, quinazolinamines **22** and **33** showed high docking scores when docked with human BCRP and/or the P-gp model, indicating the high affinity between **22**, **33** and BCRP and/or P-gp protein. The interactions between **22** and **33** and the important residues can be important clues for the further modification of quinazolinamine derivatives. These results give insight into the rational design of quinazolinamines reversing BCRP and P-gp-mediated MDR in cancers.

## Data Availability

Not applicable.

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
