# Peer review of "Design, Synthesis and Biological Evaluation of Quinazolinamine Derivatives as Breast Cancer Resistance Protein and P-Glycoprotein Inhibitors with Improved Metabolic Stability"

_biomolecules, 2023, doi:10.3390/biom13020253_

Round 1

Reviewer 1 Report

derivatives. The manuscript is well written, and the experimental data are well described. In general abbreviations should not be used in titles or in abstracts, please write full names. I also find the abbreviation list a little un-nessesary: TLC and organic compounds, such as THF written in experimental part needs no abbreviation, however in Introduction and in Results and discussion full names should be written for clarity. Abbreviations should follow in parentheses. Headlines such as “Chemistry” must be avoided, please write full description of what the paragraph is about. Also include compounds of which 1H NMR and 13C NMR spectra were recorded. Product purity should not be given in general, must be given for each compound in addition to grams and moles. Compound numbers must be in bold, check whole manuscript.

Author Response

Thank you so much for your suggestions. The manuscript has been revised according to your advice.

Reviewer 2 Report

In this manuscript, the authors report synthesis of structurally diverse quinazolinamine derivatives starting from readily available precursors. Final step is a nucleophilic substitution of corresponding chloroquinazolines with a series of amines. Some of the final products containing an amino group were converted to several new derivatives (30-33). Core structures of these heterocyclic compounds are known to demonstrate potent inhibitory activities on BCRP and P-glycoprotein (P-gp). Some compounds showed dual BCRP and P-gp inhibitor activities, and some only BCRP (e.g., 22 and 33). Compounds 22 and 33 were subjected to mechanistic studies. In addition, dual BCRP and P-gp inhibitors showed improved metabolic stability than Ko143 - structural analog of FTC.

The work is clearly presented in terms of synthesis, biological studies on BCRP and P-gp inhibition, and molecular docking. I believe it should be interesting for readers.

My concerns and suggestions are the following:

Major: Compounds 6-8, 9-11, 15-17 are not fully characterized: missing m.p., 13C NMR, and HRMS. The authors did not provide references if those compounds were known in literature. My search indicates that some compounds have been reported and characterized, and some have not. The authors need to do full characterization for all new compounds, or provide references for previously reported and characterized compounds.

Minor: 1. Make sure that numbers of all compounds are consistently bolded. 2. Section 2.1.4 includes general procedure for synthesis of compounds 12-14, but 15 needs to be separated.

Author Response

Indeed, Compounds 6-815-16 have been previously reported and the references were added accordingly. Even though compounds 9-11 and 17 are new compounds, they are the intermediates, not the final compounds. The intermediates were characterized with H NMR, and all the final compounds have been fully characterized. Since the first author was graduated, it is very difficult for us to add data of m.p., 13C NMR, and HRMS for the four compounds. The numbers of all compounds are consistently bolded in our revised version. The procedure of compound 15 was separated in the revised version.

Reviewer 3 Report

Chao-Yu Cai and coworkers present the description of a series of a number of Quinazolinamine derivatives able to interact with two transporter proteins. The paper is well conducted and I have no objection  for publication except it would be better, for a wider audience, to target some pharmaceutical journals instead of Biomolecules.
The manuscript is well written and clear. The only comment I have would be to reverse the experimental and results sections  for clarity and to know how the authors "prepare" the PDB files in the docking calculations (lines 490-495). Is it just removing the inhibitor ? the anti-gene fragment? what about water molecules?  Also recall that 6FN1 is not exactly a pure human protein.
All in all, a good work

Author Response

Thank you so much for your positive evaluation and comments to improve the manuscript. The procedures of preparing the protein for molecular modelling has been included. The preparation of protein follows the default protocol. The inhibitor was removed; hydrogens were added; the disulfide bonds were created; the waters were removed; restrained minimization was conducted. Indeed, P-gp model 6FN1 is a human-mouse chimeric model. The human P-gp model (PDB ID: 7A6E) is more appropriate to investigate the interactions of P-gp with compound 22 and verapamil. The results and discussion have been revised.

Round 2

Reviewer 2 Report

It is unfortunate that the authors could not provide full characterization for all new compounds even they are intermediates. But I am satisfied with their responses and revisions. The paper can now be published.